# High-Risk Biliary Anastomosis During Robotic Pancreaticoduodenectomy: Initial Experience with Biodegradable Biliary Stent

**DOI:** 10.3390/medicina60111798

**Published:** 2024-11-01

**Authors:** Carolina González-Abós, Claudia Lorenzo, Samuel Rey, Francisco Salgado, Fabio Ausania

**Affiliations:** 1Hepatibiliopancreatic and Transplant Department, General and Digestive Surgery, Hospital Clinic, University of Barcelona, C. Villarroel, 170, 08036 Barcelona, Spain; cagonz@clinic.cat (C.G.-A.); clorenzo@clinic.cat (C.L.); samuelreyrobledo@gmail.com (S.R.); fsalgado@clinic.cat (F.S.); 2Institut d’Investigacions Biomèdiques August Pi i Sunyer (IDIBAPS), 08036 Barcelona, Spain; 3Department of Surgery, University of Barcelona, C. Casanova, 143, 08036 Barcelona, Spain

**Keywords:** robotic pancreaticoduodenectomy, biodegradable stents, biliary fistula

## Abstract

*Background and Objectives*: Biliary fistulas (BFs) occur in approximately 3–8% of patients undergoing pancreaticoduodenectomy (PD), and the bile duct diameter ≤ 5 mm is the most important risk factor. The aim of this study was to evaluate the efficacy of biodegradable biliary stents (BSs) in reducing complications in patients undergoing robotic pancreaticoduodenectomy (RPD) with a bile duct diameter of ≤5 mm. *Materials and Methods*: A retrospective single-centre observational study was conducted. Patients undergoing RPD after the completion of the robotic biliary anastomosis learning curve were included in this study. Only patients with a bile duct diameter ≤ 5 mm were included in the analysis. A prospectively held database was used. The intraoperative time for biliary anastomosis was extracted from surgical videos. *Results*: Of 30 patients, 20 received no biliary stent (nBS) and 10 received a biodegradable stent (BS). The decision to use a stent was based on product availability. The median operative time for biliary anastomosis was significantly shorter in the BS group compared to the nBS group, at 15 min versus 24 min (*p* < 0.001). Three patients in the nBS group developed a BF, whereas none were observed in the BS group. No stent migration was observed in any of the patients. *Conclusions*: The use of biodegradable biliary stents in high-risk biliary anastomosis in RPD appears to effectively reduce the incidence of BFs and may serve as a viable strategy to mitigate early biliary complications. The use of biodegradable stents facilitates a faster and easier biliary anastomosis. These findings suggest a potential benefit of using biodegradable stents in complex biliary reconstruction. However, larger studies are needed to confirm these results.

## 1. Introduction

Pancreaticoduodenectomy (PD), also known as the Whipple procedure, is the most commonly performed surgery for the treatment of malignant or premalignant lesions located in the periampullary region, which includes the pancreas, bile duct, duodenum, and surrounding structures [1]. This complex and demanding surgical procedure involves the resection of the pancreatic head, duodenum, a portion of the bile duct, the gallbladder, and sometimes part of the stomach. Reconstruction is achieved through three critical anastomoses: the bilioenteric anastomosis (connecting the bile duct to the intestine), the pancreatic anastomosis (connecting the pancreatic remnant to the intestine), and the gastroenteric anastomosis (connecting the stomach to the intestine) [2]. Given its intricacy, PD carries a high risk of complications, including infection, bleeding, and anastomotic leaks [3].

In recent years, robotic pancreaticoduodenectomy (RPD) has emerged as a promising alternative to traditional open surgery [4]. The adoption of robotic systems in PD allows surgeons to perform the procedure with enhanced precision, flexibility, and control, through minimally invasive techniques. This technological advancement has “democratised” the ability to perform PD, making it more accessible to surgeons trained in minimally invasive approaches [5]. However, RPD is still considered a highly specialised procedure, and should only be performed in high-volume centres with extensive expertise in both pancreatic surgery and robotic techniques, to ensure patient safety and optimal outcomes [6].

One of the notable complications following PD is the development of a biliary fistula (BF), which occurs in approximately 3–8% of patients [7]. A BF is characterised by the leakage of bile from the surgical connection between the bile duct and the intestine, often associated with a small-diameter common bile duct (CBD) [7]. The risk of mortality significantly increases when a BF is accompanied by a postoperative pancreatic fistula (POPF), another serious complication where pancreatic fluid leaks from the pancreatic anastomosis [8]. These complications can lead to severe infections, prolonged hospital stays, and increased healthcare costs.

The advent of robotic surgery has facilitated the execution of complex PD cases, including those with challenging biliary and pancreatic anastomoses [6]. Despite these advancements, there remains a lack of established strategies to mitigate biliary complications in technically difficult anastomoses during robotic surgery. Some surgical teams have begun experimenting with the use of free stents in hepaticojejunostomy, a technique where a stent is placed in the connection between the liver bile duct and the intestine, to reduce bile leakage and simplify the anastomosis process [9]. However, the insertion of non-extractable foreign materials in the bile duct is generally discouraged due to the risk of long-term complications, such as stent migration, infection, and bile duct obstruction.

The aim of this study is to analyse the feasibility, safety, and efficacy of biodegradable stents during hepaticojejunostomy in robotic pancreaticoduodenectomy (RPD) in patients with a small bile duct (≤5 mm). By evaluating a cohort of patients undergoing RPD with biliary stenting, we aim to assess whether this technique can effectively reduce the incidence of biliary complications, improve anastomotic integrity, and improve the overall surgical outcomes. The results of this study may provide valuable insights into optimising surgical techniques in robotic PD and improving patient care.

## 2. Materials and Methods

All consecutive patients who underwent robotic pancreaticoduodenectomy (RPD) at our centre were included in this observational retrospective study. All data were extracted from a prospectively held database. The inclusion criteria were as follows: any indication for pancreaticoduodenectomy, a robotic surgical approach, a non-dilated bile duct on preoperative imaging, and an age over 18 years. Patients with preoperative biliary stenting were excluded, due to bile duct manipulation and bile duct dilation. To ensure surgeon proficiency, only patients operated on after the surgeon (FA) had completed 20 robotic biliary anastomoses were included in the analysis. The decision to use a stent was based on its availability.

The primary outcome was the comparison of early postoperative biliary complications between RPD patients with biodegradable internal biliary stents and those without stents in hepaticojejunostomy. Secondary outcomes included the following: (a) comparing the time required to perform hepaticojejunostomy between the two groups and (b) comparing long-term biliary complications in patients undergoing RPD with biodegradable internal biliary stents versus no stents in hepaticojejunostomy.

The ARCHIMEDES™ Biodegradable Biliary and Pancreatic Stent (AMG International GmbH, Bochstrasse 16, 21423 Winsen, Germany) features a helical design that permits bile or pancreatic juice to flow along the outer extremity while maintaining duct patency. This stent is made of Polydioxanone, Polyethylene glycol, and Barium Sulphate. The stent can be placed during an endoscopic procedure or surgery involving biliary anastomosis. However, there are no previous reports on its use during robotic pancreaticoduodenectomy; therefore, no adverse events have been reported related to the specific use assessed in this manuscript. The full degradation of these stents is expected after 6 weeks. The device requires no specific training or storage conditions, is intended for single use, and cannot be re-sterilised. In patients selected for stent placement, the stent was inserted into the hepaticojejunostomy after creating the posterior wall of the anastomosis. Anastomoses were performed using two semi-continuous layers of absorbable 4-0 barbed suture. The biliary anastomosis was always conducted after the pancreaticojejunostomy.

Surgical videos were routinely recorded and reviewed for this study. The following metrics were evaluated:Time spent performing the hepaticojejunostomy;Presence of intraoperative bile leaks following the anastomosis;Need for hepaticojejunostomy revision (if an intraoperative bile leak was detected);Perceived difficulty of stent placement.

Clinically relevant bile leaks were defined according to the International Study Group of Liver Surgery criteria. Pure bile leaks were also identified based on the co-occurrence of pancreatic fistula.

The postoperative period was managed as previously described. Following hospital discharge, all patients were followed up clinically at 1 month, and then every 6 months with abdominal imaging (CT) and blood tests, including liver function tests.

Data analysing the impact of the use of biodegradable stents were compared. SPSS version 26.0 (IBM, Armonk, NY, USA) was used for data management and analysis. Quantitative variables were described as means ± standard deviations, and qualitative variables were described as absolute frequencies and percentages. Other continuous data were expressed as the median values with a range using the Wilcoxon rank-sum test. For two-sided group comparisons, the chi-square test or Fisher’s exact test were used for categorical variables and the Mann–Whitney U test was applied for continuous variables. A value of *p* < 0.05 was considered statistically significant.

This research was conducted in accordance with the protocol, the principles of the latest revised version of the Declaration of Helsinki (Seoul, 2008), the standards of Good Clinical Practice as outlined by the Harmonized Tripartite Standards of the ICH for Good Clinical Practice (1996), and the guidelines for Good Epidemiological Practice (URL: http://www.ieaweb.org accessed on 10 June 2024). This study was conducted in accordance with the Declaration of Helsinki and approved by the Comité de Ética de la Investigación con medicamentos (CEIm) (Approval Code: HCB/2024/0924, Approval Date: 10 March 2024).

## 3. Results

Patients’ characteristics are shown in Table 1. There were no patients with preoperative biliary drainage.

A total of 85 patients underwent a robotic hepaticojejunostomy during the study period. Out of 30 patients matching the inclusion criteria, 20 patients did not receive a biliary stent (nBS), and 10 patients received a BS. Stent placement was decided based on the availability of the product. The median age was 68 years old, 57.7% of the patients were male, and 36.7% were high-risk patients according to the PD-Roboscore classification.

Postoperative outcomes are shown in Table 2.

Results on primary and secondary outcomes are shown in Table 3. The median operative time for biliary anastomosis was 22 min (15 vs. 24 min; *p* < 0.001). In two patients of the nBS groups, temporal biliary stents for facilitating anterior wall anastomosis were used (Figure 1). Three patients developed a clinically relevant pure biliary fistula in the nBS group, and there were no biliary fistulas in the BS group. Two patients were treated by conservative management and one patient underwent a laparotomy and re-hepaticojejunostomy. No stent migration was observed. In two cases, an intraoperative re-hepaticojejunostomy was performed, due to the detection of a biliary leak. There was no mortality in this series. No cases of stent migration were observed. No adverse events were reported associated with the use of a BS. No postoperative biliary stent-associated infections were reported in these patients. Surgeon perception was that using biliary stents makes the biliary anastomosis easier, as there is no risk of including the posterior wall when performing the anterior wall suture, and a minimum calibre of the anastomosis can be guaranteed, mitigating the possibility of an anastomotic stricture in the short- and long-term outcome (Figure 1 and Figure 2). 

The median follow-up time was 12 (6–15) months. All patients received a CT scan at 6 months. There were no biliary strictures or complications related to stent migration during follow up. Further follow-ups were performed with a CT scan or magnetic resonance according to the patient’s initial indication for pancreaticoduodenectomy.

## 4. Discussion

This is the first study to explore the feasibility of using biodegradable biliary stents during minimally invasive pancreaticoduodenectomy (PD). In our preliminary experience, the use of biodegradable biliary stents during robotic pancreaticoduodenectomy (RPD) showed very promising results. Notably, there were no biliary fistulas (BFs) in cases where the biliary stent was used, and no reoperations were required, either intraoperatively or postoperatively.

According to the International Study Group on Liver Surgery (ISGILS), a biliary leak is defined as a biliary fluid bilirubin concentration that is three times the plasma bilirubin concentration on postoperative day (POD) 3 [10]. The BF is then graded according to the severity of the leak: Grade A involves no significant deviation from standard postoperative care, Grade B requires additional radiological and pharmacological treatment, and Grade C necessitates surgical intervention. An incidence of up to 8% has been reported for BFs following pancreaticoduodenectomy [6]. Biliary leak onset usually occurs on postoperative day 2–3. Compared to the postoperative pancreatic fistula (POPF), the BF has received less attention due to its lower frequency, morbidity, and mortality rates [3]. However, a BF can lead to bile peritonitis and abdominal abscesses, which can prolong hospital stays and increase mortality rates. Mortality is particularly high when a BF is associated with a POPF. Previous studies have shown that nearly half of the patients with a BF also have an associated POPF, with major morbidity and mortality rates reaching 64% and 19%, respectively [11]. The endoscopic treatment of early biliary complications is usually not performed, as early endoscopic procedures might increase the risk of anastomotic failure and cause severe pneumoperitoneum [12]. Endoscopic procedures are typically reserved for treating late biliary complications, when feasible [12]. The conservative treatment of BFs is often successful, although the decision to perform an early redo of hepaticojejunostomy or a percutaneous approach depends on the patient’s condition, surgeon expertise, and local resource availability. Known risk factors for BFs include excessive skeletonization of the hepatic duct, a small ductal diameter, and anastomosis to the common bile duct rather than the common hepatic duct [7]. Many protective measures have been attempted with little success, including various reconstruction techniques and intraoperative T-tube placement [13]. The preoperative prediction of BFs may be advantageous for early treatment, but it has not been extensively studied for PD. There are no studies showing the intraoperative use of biodegradable stents in hepaticojejunostomy, while some publications suggest that the use of these stents could reduce complications of the pancreatic anastomosis during open pancreaticoduodenectomy [14,15].

Over the past few decades, pancreatic surgery has made tremendous progress. The introduction of minimally invasive techniques, particularly laparoscopy and robotic platforms, has revolutionised the performance of complex pancreatic surgeries. Pancreaticoduodenectomy remains one of the most technically challenging surgeries. Early attempts to improve surgical outcomes with laparoscopy were limited until the advent of robotic surgery, which overcomes many limitations of laparoscopy, such as the fulcrum effect, fine instrument manipulation, and the reversal of instrument tip movement. Several studies have reported the feasibility and safety of robotic PD in treating pancreatic head malignancies, following the first case reported by Giulianotti et al. in 2003 [16]. Previous reviews and meta-analyses have shown that perioperative outcomes of robotic PD are at least equivalent to those of open PD [17,18,19]. While short-term clinical outcomes of robotic PD have been well studied, long-term results are less documented due to the limited longitudinal data. Bile leak rates following RPD can be particularly high during the learning curve, up to 23% [20]. Even data from randomised controlled trials following the learning curve are not encouraging. Hackert et al. reported a Grade B/C biliary leak rate of 17.2% for RPD compared to 9.1% for open surgery, indicating a tendency towards more biliary complications with the robotic approach [21]. However, RPD study data are difficult to interpret, as there is no stratification according to the bile leakage risk. To address this, our study included only patients with non-dilated bile ducts (Figure 2), excluding those with preoperative biliary stenting. By focusing on this subset of patients, we aimed to better understand the impact of biodegradable stent use on biliary complications in a controlled cohort.

No adverse events were observed in patients undergoing RPD with a BS (Figure 2), and no complications related to stent degradation were reported. Also, no stent migration was reported; this is an important issue as there were no episodes of postoperative cholangitis, and no patients underwent an endoscopic procedure to remove the stent.

This study has several limitations. First, as an observational retrospective study, it has the typical bias associated with this study design. However, the cases were collected from a prospective database, and a surgical video review allowed the accurate assessment of intraoperative events. Secondly, the sample size was small; however, this was a preliminary study aimed at assessing the feasibility, safety, and efficacy of biodegradable stents in a specific setting. Additional factors that could have helped to better stratify the risk of a biliary fistula, such as biliary tree vascularisation and bacterial colonisation of the bile, could not be analysed. In this sense, the role of biliary stenting in patients with preoperative biliary dilatation or preoperative biliary stenting needs to be evaluated in future studies. Long-term follow-up data are also not available, which limits the assessment of long-term effects on biliary strictures and the estimation of a cost analysis. In addition, the single-centre design of this study may limit the generalisability of the findings to other settings. However, to confirm these findings and to establish the long-term safety and efficacy of biodegradable biliary stents in this setting, larger, multicentre studies with longer follow-up times are needed. In addition, to better understand the economic implications of this approach, future studies should consider stratifying patients based on the risk of bile leakage, and incorporating cost analyses.

In conclusion, early experience with biodegradable biliary stents during robotic pancreaticoduodenectomy has shown promising results, with no biliary fistulas observed in the stent group and no need for re-operation. These findings suggest that the use of biodegradable biliary stents may be a valuable addition to the surgical management of RPD patients, potentially reducing the frequency of biliary complications and improving patient outcomes.

## Figures and Tables

**Figure 1 medicina-60-01798-f001:**
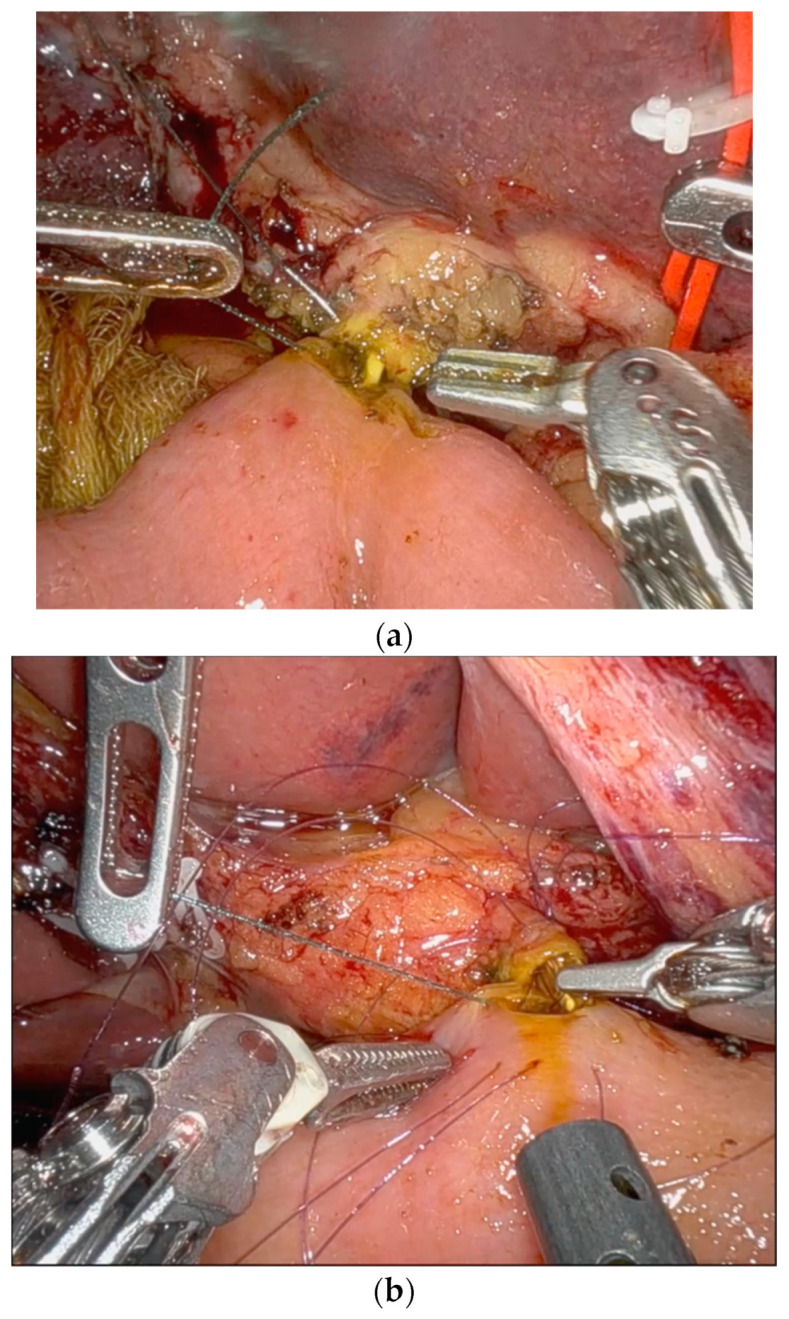
(**a**) Biliary duct < 5 mm during pancreaticoduodenectomy. This image shows the ability to perform an anterior running suture thanks to the use of a biodegradable stent. (**b**) Biliary duct < 5 mm during pancreaticoduodenectomy. This image shows the need to perform an interrupted suture due to the small size of the BD and the use of a non-biodegradable stent to protect the posterior wall. The non-biodegradable stent was intraoperatively removed after passing all the sutures of the anterior wall.

**Figure 2 medicina-60-01798-f002:**
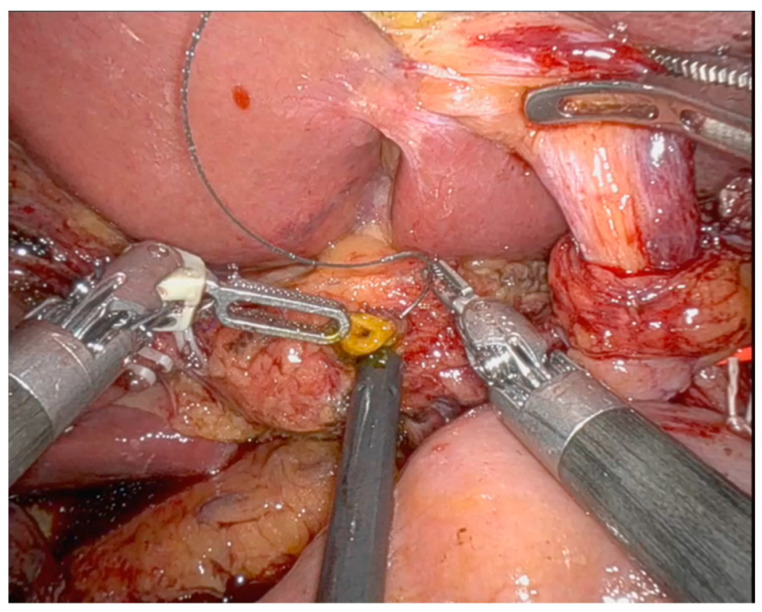
Biliary duct < 5 mm during pancreaticoduodenectomy. This image shows the start of the hepaticojejunostomy performance.

**Table 1 medicina-60-01798-t001:** Patient baseline features. IQR, interquartile range; BMI, body mass index; ASA, American Society.

	Biliary Stent (n = 10)	No Biliary Stent (n = 20)	*p*-Value
Age (years), mean (IQR)	68 (55–73)	69 (54–78)	0.826
Sex ratio (male), n (%)	7 (70)	12 (60)	0.287
BMI, mean (IQR)	26 (22–34)	27 (23–35)	0.816
ASA score, n (%)			0.794
II	4 (40)	9 (45)	
III	6 (60)	11 (55)	
Smoking, n (%)			0. 7290
Present	1 (10)	1 (5)	
Past	1 (10)	3 (15)	
Never	8 (80)	16 (80)	
Alcohol consumption, n (%)			0.604
Yes, often	1 (10)	2 (10)	
Yes, occasionally	1 (10)	3 (15)	
Yes, rarely	6 (10)	10 (50)	
Past	1 (10)	3 (15)	
Never	1 (10)	1 (5)	
Unknown	0 (10)	1 (5)	
Previous abdominal surgeries, n (%)	6 (60)	11 (55)	0.954

**Table 2 medicina-60-01798-t002:** Short-term postoperative outcomes. CCI, comprehensive complication index; ICU, intensive care unit; SD, standard deviation; SSI, surgical site infection; POPF, postoperative pancreatic fistula; PPH, post-pancreatectomy haemorrhage; DGE, delayed gastric emptying.

	Biliary Stent (n = 10)	No Biliary Stent (n = 20)	*p*
CCI	8 (11.4)	15 (14.7)	0.080
Length of stay in the ICU (days), mean (SD)	1 (1)	1 (2)	0.433
Length of hospital stay (days), mean (IQR)	10 (3)	11 (10)	0.284
Superficial SSI within 30 days, n (%)	1 (10)	3 (15)	0.704
POPF grade B/C, n (%)	3 (30)	5 (25)	0.773
PPH grade B/C, n (%)	0 (0)	1 (5)	0.472
DGE grade B/C, n (%)	2 (20)	6 (30)	0.553
Chyle leak grade B/C, n (%)	0 (0)	0 (0)	
Number of patients with re-operation(s), n (%)	0 (0)	2 (10)	0.300
Number of patients with readmission(s), n (%)	1 (10)	3 (15)	0.704

**Table 3 medicina-60-01798-t003:** Comparison of perioperative outcomes in patients with BS vs. patients without BS. NS: not statistically significant.

	Biliary Stent (n = 10)	No Biliary Stent (n = 20)	*p* Value
Time for biliary anastomosis, minutes (range)	15 (13–18)	24 (20–32)	<0.001
Intraoperative re-do	0	2	NS
Postoperative biliary fistula, grade B/C (%)	0	3 (15)	NS
Anastomotic biliary stricture	0	0	NS

## Data Availability

Dataset available on request from the authors.

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
