# Peer review of "High-Risk Biliary Anastomosis During Robotic Pancreaticoduodenectomy: Initial Experience with Biodegradable Biliary Stent"

_medicina, 2024, doi:10.3390/medicina60111798_

Round 1
Reviewer 1 Report
Comments and Suggestions for Authors
The manuscript with the title "High-risk biliary anastomosis during robotic pancreaticoduodenectomy: initial experience with biodegradable biliary stent" aims to evaluate the usefulness, efficacy, safety, and results of using biodegradable biliary stents when performing robotic bilioenteric anastomosis (hepaticojejunostomy) during a robotic pancreaticoduodenectomy.
The manuscript is interesting, and brings a novel concept for biliary anastomoses performance, although it has a major limitation in the small cohort of patients.
There are some aspects that should deserve clarification:
-in table 1, excepting age, indicate what the numbers in the round brackets represent;
-in the discussions: the authors indicate that the study was retrospective, while in the methods they state that it was prospective;
-better delimit the conclusions;
-some details regarding the type of material used for the biodegradable stents, variants existing on the market currently, average price, adverse effects/complications reported for such implants, intolerance cases would be beneficial;
-which is the time interval required for full biodegradation;
- more reports in the discussions regarding other studies where biodegradable stents were used ( other anastomotic types);
-putative uses of such implants in open surgery/other types of operations.
Author Response
Comment 1: In table 1, excepting age, indicate what the numbers in the round brackets represent.
Thank you for your comment. The manuscript was edited accordingly.
Comment 2: in the discussions: the authors indicate that the study was retrospective, while in the methods they state that it was prospective;
Thank you for your comment. The study was retrospective, on a prospectively held database. This was now clarified in the methods paragraph as well.
Comment 3: better delimit the conclusions;
Conclusions paragraph has been summarized.
Comment 4: some details regarding the type of material used for the biodegradable stents, variants existing on the market currently, average price, adverse effects/complications reported for such implants, intolerance cases would be beneficial;
Thank you for your comment. This was added to the methods paragraph.
Comment 5: which is the time interval required for full biodegradation;
Thank you for your comment. Time for full degradation of this type of stents is 6 weeks. This was described in the methods paragraph.
Comment 6: more reports in the discussions regarding other studies where biodegradable stents were used (other anastomotic types);
Thank you for your comment. We have added the very limited previous experience on the use of BS in pancreatic anastomosis in the discussion paragraph.
Comment 7: putative uses of such implants in open surgery/other types of operations.
Thank you for your comment. The use of this implants was tested in open pancreatic surgery. The manuscript was edited accordingly.
Reviewer 2 Report
Comments and Suggestions for Authors
Interesting article, here my comment's:
- In the methods section it's reported that the anastomoses have been performed using a two semi-continuous layers of absorbable 4-0 barbed suture but in Figure 2b is represented an interrupted suture. Need to be specify why and how many have been performed in the two methods.
- Three BL have been reported in the group without the reabsorbable stent. It's a 15% of BL in that group. It's a quite high % even if considering the risk factors. What are your considerations about?
- What about the cost of the stent?
- Just to know if you have the percentage of a concomitant pancreatic fistula
- The comparison between with and without the stent is promising but the serie is quite small (as well specified in the limitations by the author).
Comments on the Quality of English Language
Minor revision
Author Response
Comment 1: In the methods section it's reported that the anastomoses have been performed using a two semi-continuous layers of absorbable 4-0 barbed suture but in Figure 2b is represented an interrupted suture. Need to be specify why and how many have been performed in the two methods.
Thank you for your comment. All anastomosis where a biodegradable stent was placed have been performed using two semi-continuous layers of absorbable 4-0 barbed suture. Figure 2b aimed to show the need to perform an interrupted suture in case of having a non-biodegradable stent that should not remain inside the biliary anastomosis.
Comment 2: Three BL have been reported in the group without the reabsorbable stent. It's a 15% of BL in that group. It's a quite high % even if considering the risk factors. What are your considerations about?
Thanks for your interesting question. Bile leak following robotic PD, is quite high. For instance, in the worldwide highest volume pancreatic surgery center (Heidelberg University), the biliary fistula rate, in non-high-risk patients is 17% (Klotz R, et Al. Robotic versus open partial pancreatoduodenectomy (EUROPA): a randomised controlled stage 2b trial. Lancet Reg Health Eur. 2024 Feb 22;39:100864.) compared to open surgery (which is 9% anyhow). We believe that 15% is an acceptable complication rate considering the very high-risk population selected for out study.
Comment 3: What about the cost of the stent? Thanks for your comment. The stent cost has a variable price because it depends on the policy of the Institution, the country, the number of stents used annually, etc. We are now performing a cost-analysis study, on a larger population, and we will publish these data soon. For the current study, we used free samples, therefore I am afraid we are not able to describe the real costs.
Comment 4: Just to know if you have the percentage of a concomitant pancreatic fistula
In this cohort, one of the three patients presenting biliary fistula developed also a grade B pancreatic fistula. Mortality in this cohort was nil.
Reviewer 3 Report
Comments and Suggestions for Authors
Congratulations to the authors; this is a topical article whose subject is generating much discussion in the literature
However, the article needs improvement, especially in terms of material and method, where various inconsistencies appear.
For example - in the introduction the study is described as an observational study; then in the method chapter it is not explained what type of study the authors are conducting and finally in the discussion of the study's limitations it is described as a retrospective study.
Also, in the method chapter, the inclusion and exclusion criteria need to be explained more clearly and the statistical method used to interpret the data needs to be added.
Informed consent cannot be waived, as the authors explain, even if the data are anonymized (they show intraoperative images).
Another inconsistency arises in the postoperative follow-up of patients - initially, the authors say that follow-up is done with CT and MRI, but in the discussion, follow-up is done with CT only
The images (fig1,2a,2b) should be moved to discussion.
Fig 2a shows the temporary use of a non-biodegradable stent, but no details of the number of patients in which it was used... results and discussion of this should be added.
I strongly recommend reanalyzing the material and method chapter and of course, the discussions should be made following them.
Comments on the Quality of English Languageminor english editing required
Author Response
Comment 1: However, the article needs improvement, especially in terms of material and method, where various inconsistencies appear.
For example - in the introduction the study is described as an observational study; then in the method chapter it is not explained what type of study the authors are conducting and finally in the discussion of the study's limitations it is described as a retrospective study.
Thanks for your comments. We clarified that this is a retrospective observational study, although the data were extracted from a prospectively held database.
Also, in the method chapter, the inclusion and exclusion criteria need to be explained more clearly, and the statistical method used to interpret the data needs to be added.
Thanks for your comments. We described the inclusion criteria in the methods paragraph: any indication for pancreaticoduodenectomy, a robotic surgical approach, non-dilated bile duct on preoperative imaging, and age over 18 years. The exclusion criteria were: Patients with preoperative biliary stent were excluded
However, please let us know what type of criteria you think is missing and we will be happy to include it.
Statistical analysis methodology was added according to your suggestion.
Informed consent cannot be waived, as the authors explain, even if the data are anonymized (they show intraoperative images).
Thanks for your comment. We think there might be a misunderstanding between informed consent to use the stents and informed consent to publish the data.
Regarding the first issue, patients were not asked for a specific consent since this was not a prospective study with a deviation from standard treatment; however, before the operation, all patients must sign a consent form explaining that intraoperative devices might be used if considered of any benefit for the patient. For instance, in many cases of high-risk biliary or pancreatic anastomosis, we would perform an external drainage, although there is no specific consent for it. Please note that in our Country, this stent is not considered a medication as it is only a medical device; and also, this is not a permanent device, and it has European approval for use in the bile duct.
Regarding the second issue, we do have approval of the Ethical Committee to publish the data, which is now included in the Method paragraph. Also, all patients included in the study had previously signed an authorization to use the data and the images for research purposes following anonymization.
We have now clarified these two issues in the manuscript.
Another inconsistency arises in the postoperative follow-up of patients - initially, the authors say that follow-up is done with CT and MRI, but in the discussion, follow-up is done with CT only
Thanks for your comment. We included the MRI because some patients might have allergy to contrast medium or a benign disease. However, all patients included ended up with having a CT scan, and therefore there was a difference between the methods and the discussion. We have now eliminated the MRI according to your suggestion.
The images (fig1,2a,2b) should be moved to discussion.
Thanks for your comment. We moved the figures to the Discussion paragraph.
Fig 2a shows the temporary use of a non-biodegradable stent, but no details of the number of patients in which it was used... results and discussion of this should be added.
Thanks for your comment. Non-biodegradable stents were not used in this study, as described in the results. We used this picture to explain our alternative in case non-biodegradable stent is not available. However, if you think this picture is confusing, we would be happy to remove it.
I strongly recommend reanalyzing the material and method chapter and of course, the discussions should be made following them.
The manuscript was edited according to your suggestions.
Comments on the Quality of English Language
Minor English editing required
Thanks for your comment. We have further reviewed the manuscript with a mother tongue English language speaker.
Round 2
Reviewer 3 Report
Comments and Suggestions for Authors
Congratulations to the authors,
The article was greatly improved both from scientific and ethical point of view and I am glad that I was able to participate in this aspect